# Self-supervised Point Cloud Prediction
# Using 3D Spatio-temporal Convolutional Networks

**Benedikt Mersch**   **Xieyuanli Chen**   **Jens Behley**   **Cyrill Stachniss**
University of Bonn
{mersch, xieyuanli.chen, jens.behley, cyrill.stachniss}@igg.uni-bonn.de

**Abstract:** Exploiting past 3D LiDAR scans to predict future point clouds is a promising method for autonomous mobile systems to realize foresighted state estimation, collision avoidance, and planning. In this paper, we address the problem of predicting future 3D LiDAR point clouds given a sequence of past LiDAR scans. Estimating the future scene on the sensor level does not require any preceding steps as in localization or tracking systems and can be trained self-supervised. We propose an end-to-end approach that exploits a 2D range image representation of each 3D LiDAR scan and concatenates a sequence of range images to obtain a 3D tensor. Based on such tensors, we develop an encoder-decoder architecture using 3D convolutions to jointly aggregate spatial and temporal information of the scene and to predict the future 3D point clouds. We evaluate our method on multiple datasets and the experimental results suggest that our method outperforms existing point cloud prediction architectures and generalizes well to new, unseen environments without additional fine-tuning. Our method operates online and is faster than the common LiDAR frame rate of 10 Hz.

## 1   Introduction

Sequential 3D point clouds obtained from Light Detection And Ranging (LiDAR) sensors are widely used for numerous tasks in robotics and autonomous driving, like localization [1], detection [2, 3], mapping [4], segmentation [5, 6], and trajectory prediction [7]. The ability to forecast what the sensor is likely to see in the future can enhance decision-making for an autonomous vehicle. A promising application is to use the predicted point clouds for path planning tasks like collision avoidance. In contrast to most approaches, which predict for example future 3D bounding boxes of traffic agents, point cloud prediction does not need any preceding inference steps such as localization, detection, or tracking to predict a future scene. Running an off-the-shelf detection and tracking system on the predicted point clouds yields future 3D object bounding boxes as demonstrated by recent approaches for point cloud forecasting [8, 9]. From a machine learning perspective, point cloud prediction is an interesting problem since the ground truth data is always given by the next incoming LiDAR scans. Thus, we can train our point cloud prediction self-supervised without the need for expensive labeling and evaluate its performance online in unknown environments.

In this paper, we address the problem of predicting large and unordered future point clouds from a given sequence of past scans as shown in Fig. 1. High dimensional and sparse 3D point cloud data render point cloud prediction a challenging problem that is not yet fully explored in the literature. A future point cloud can be estimated by applying a predicted future scene flow to the last received scan or by generating a new set of future points. In this paper, we focus on the generation of new point clouds to predict the future scene. In contrast to existing approaches [8, 9], which exploit recurrent neural networks for modeling temporal correspondences, we use 3D convolutions to jointly encode the spatial and temporal information. Our proposed approach takes a new 3D representation based on concatenated range images as input and jointly estimates a future range image and per-point scores for being a valid or an invalid point for multiple future time steps. This allows us to predict detailed future point clouds of varying sizes with a reduced number of parameters to optimize resulting in faster training and inference times. Furthermore, our approach is also fully self-supervised and does not require any manual labeling of the data.

5th Conference on Robot Learning (CoRL 2021), London, UK.

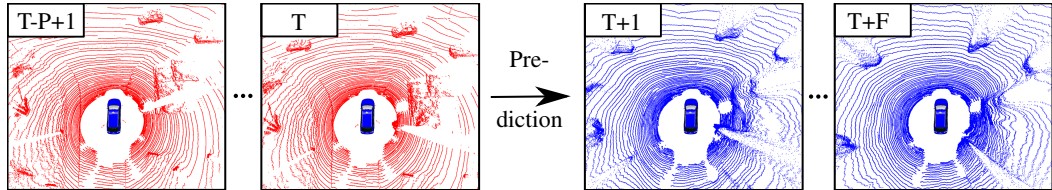

Figure 1: Given a sequence of $P$ past point clouds (left in red) at time $T$, the goal is to predict the $F$ future scans (right in blue). Note that all point clouds are in the sensor's coordinate system. Best viewed in color.

The main contribution of this paper is a novel range image-based encoder-decoder neural network to jointly process spatio-temporal information from points clouds using 3D convolutions. Our method can obtain structural details of the environment by using skip connections and horizontal consistency using circular padding, and in the end, provides more accurate predictions than other state-of-the-art approaches for point cloud prediction. We make three main key claims: Our approach is able to (i) predict a sequence of detailed future 3D point clouds from a given input sequence by a fast joint spatio-temporal point cloud processing using temporal 3D convolutional networks (CNNs), (ii) outperform state-of-the-art point cloud prediction approaches, (iii) generalize well to unseen environments, and operate online faster than a typical rotating 3D LiDAR sensor frame rate (10 Hz). These claims are backed up by the paper and our experimental evaluation. The open-source code and the trained models are available at https://github.com/PRBonn/point-cloud-prediction.

## 2 Related Work

Extracting spatio-temporal information from point cloud sequences has been exploited in the literature for different tasks related to point cloud prediction. Liu et al. [10] extend PointNet++ [6], which extracts spatial features from a single point cloud, and propose FlowNet3D for estimating the 3D motion field between two point clouds. FlowNet3D is used by Lu et al. [11] to interpolate intermediate frames between two LiDAR scans. Other deep learning-based scene flow estimation methods can be found in the literature [12, 13]. In contrast to scene flow between two past frames, point cloud prediction estimates future point clouds. In addition to that, scene flow requires labels like point-wise correspondences or segmentation masks. This makes learning costly. In contrast to that, we focus on the self-supervised generation of new, future point clouds.

Predicting future trajectories of traffic participants based on their past motion is one possible application of LiDAR-based future scene prediction. Given LiDAR scans only, these methods rely on preceding detection and tracking modules to infer past trajectories. For example, Luo et al. [7] develop a 3D convolutional architecture based on a bird's eye view (BEV) of voxelized LiDAR scans transformed by the sensor's ego-motion. Inspired by this, Casas et al. [14] increase the prediction horizon with IntentNet and also estimate a high-level driver behavior from HD maps. Liang et al. [15] integrate a tracking module into the end-to-end pipeline to improve the temporal consistency of the predictions while Meyer et al. [3] and Laddha et al. [16] explore the range image representation to predict future 3D bounding boxes. A second method to estimate a future scene representation without the need for prior object detections is prediction performed at the sensor level. Hoermann et al. [17] directly predict a future occupancy grid map from a sequence of past 3D LiDAR scans and Song et al. [18] predict a motion flow for indoor 2D LiDAR maps. Wu et al. [19] use a spatio-temporal pyramid network to estimate voxelized BEV classification and future motion maps. Recently, Toyungyernsub et al. [20] differentiate between dynamic and static cells for grid map prediction but require tracking labels to segment moving cells. In contrast to the aforementioned approaches, our method operates on the full-size point clouds without the need for voxelization. As shown by Weng et al. [8], a reversed trajectory prediction pipeline can take the point cloud predictions and detect and track objects in future scans. This makes it possible to first train the prediction of point clouds self-supervised and then use off-the-shelf detection and tracking modules to get future trajectories.

There has been a large interest in vision-based prediction. Srivastava et al. [21] propose an LSTM [22] encoder-decoder model that takes an input sequence of image patches or feature vectors and outputs the sequence of future patches or feature vectors, respectively. Shi et al. [23]

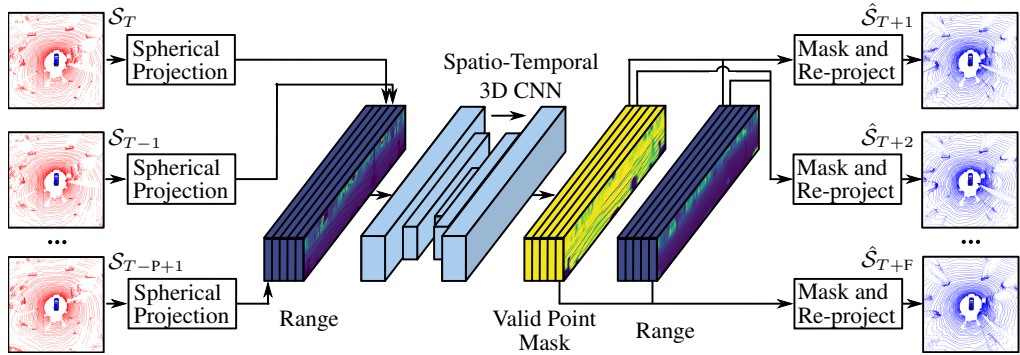

Figure 2: Overview of our approach. At time $T$, the past point clouds are first projected into a 2D range image representation and then concatenated. After passing through our proposed spatio-temporal 3D CNN network, the combined predicted mask and range tensors are re-projected to obtain the future 3D point cloud predictions.

propose ConvLSTMs that employ convolutional structures in the temporal transitions imposed by the LSTM. This makes it possible to jointly encode temporal and spatial correlations in image sequences. The integration of 3D convolutions into LSTMs has been researched by Wang et al. [24]. Aigner et al. [25] omit the use of recurrent structures and develop a video prediction approach called FutureGAN based on 3D convolutions combined with a progressively growing GAN. Using the range image representation of LiDAR scans, we can transfer the techniques from vision-based methods for solving point cloud prediction. However, there are significant differences between video prediction and range image-based point cloud prediction. The main difference is that range images contain 3D point information rather than RGB values. In addition to that, range images have a 360 degrees field of view, which causes a strong correlation between the left and right image borders because of the principle of a typical rotating LiDAR sensor, e.g., a Velodyne or an Ouster scanner.

Though vision-based and voxel-based prediction methods have been well studied, predicting full-scale future point clouds is still challenging and has not yet been fully explored in existing works. To predict a future sequence of unordered and unstructured point clouds, Fan et al. [26] propose a point recurrent neural network (PointRNN). Their method models spatio-temporal local correlations of point features and states. Lu et al. [9] introduce MoNet, which integrates a point-based context and motion encoder network and processes these features in a novel recurrent neural network. A 3D scene flow-based point cloud prediction approach has been recently developed by Deng and Zakhor [27]. All three methods are 3D point-based but only consider a limited number of points. Weng et al. [8] propose an encoder-decoder network that takes a highly compressed feature representation of a full-scale LiDAR scan and models the temporal correlations with a recurrent auto-encoder. The feature vectors are obtained either by a point-based architecture or 2D convolutional neural networks processing 2D range image representations. Deep learning-based generation of LiDAR scans from a 2D range image has been investigated by Caccia et al. [28] as well. In contrast to existing approaches, we propose a technique that concatenates the projected 2D range images to a spatio-temporal 3D representation, which makes it possible to use 3D convolutions without the need for recurrent architectures for temporal modeling. As discussed by Bai et al. [29] for general sequence modeling, using convolutional networks results in an architecture that is easier to train and faster at inference since the input range images are no longer processed sequentially. Compared to MoNet [9], this allows us to predict more future point clouds in a shorter amount of time.

## 3 Our Approach

The goal of our approach is to predict a future sequence of full-scale LiDAR points given a sequence of past scans. As illustrated in the overview of our method in Fig. 2, we first project each past 3D point cloud into a range image representation and concatenate the 2D images to obtain a 3D spatio-temporal tensor, see also Appendix A for details. Second, we pass this tensor to our encoder-decoder architecture to extract the spatial and temporal correlations with 3D convolutional kernels as described in Sec. 3.1. We use skip connections and circular padding to maintain details and horizontal consistency of the predicted range images, see Sec. 3.2. Details on the training procedure are provided in Sec. 3.3.

## 3.1 Spatio-Temporal Encoder-Decoder Architecture

Our approach first converts the 3D LiDAR points into spherical coordinates and maps them to image coordinates resulting in a dense 2D range image from which 3D information can still be recovered. If no point is projected into a specific pixel, its value is set to zero. If multiple points are mapped to the same pixel, we keep the closest point to focus on the prediction of foreground objects. For technical details on the range image projection see Appendix A. The main task of the encoder-decoder architecture illustrated in Fig. 3 is to jointly extract spatio-temporal features from the input sequence and to output future range image predictions. Compared to a multilayer perceptron architecture that disregards spatial and temporal dimensions, the use of convolutions imposes an inductive bias of having local correlations along these dimensions. Convolutional networks usually require less trainable parameters and are less prone to overfitting. When using range images for point cloud prediction with 3D convolutions, sufficient receptive fields along the spatial and temporal dimensions between encoder and decoder are required to capture the motion of points in the past frames and to propagate their locations into the future range images.

The encoder takes the 3D input tensor of size $(P, H, W)$, containing $P$ range images with height $H$ and width $W$, and first standardizes the range values based on mean and standard deviation computed from the training data. We pass the standardized tensor through a 3D convolutional input layer with $C$ kernels to get a $C$-channel feature representation of size $(C, P, H, W)$. Inspired by FutureGAN [25], the encoder block receives a tensor of size $(C_l, P_l, H_l, W_l)$ at each encoder stage $l$, applies a combination of 3D convolution, 3D batch normalization (BN) [30], and leaky ReLU activation function [31] while maintaining the size of the tensor. A strided 3D convolution follows resulting in a downsampled tensor of size $(C_{l+1}, P_l - p_l, \frac{H_l}{h_l}, \frac{W_l}{w_l})$ where $h_l$ and $w_l$ are predefined downsampling factors for the spatial feature size and $p_l$ reduces the temporal feature dimension. We batch normalize the resulting tensor and apply a leaky ReLU activation function. The kernel size is $(p_l, h_l, w_l)$ and the stride $(1, h_l, w_l)$ to achieve the desired downsampling. The downsampling compresses the sequential point cloud feature representation and forces the network to learn meaningful spatio-temporal features.

To predict future range images for $F$ future time steps, the decoder subsequently upsamples the feature tensor to the final output size $(2, F, H, W)$. Note that the number of future range images is fixed in the architecture, but longer prediction horizons can be achieved in an auto-regressive manner. This is done by sequentially feeding back the predicted range images as the input tensor. In this work, we will only focus on the prediction of a fixed number of future point clouds. The decoder architecture is a mirrored version of the encoder. First, a transposed 3D convolutional layer with kernel size $(p_l, h_l, w_l)$ and stride $(1, h_l, w_l)$ increase the feature map. We insert a 3D batch normalization and leaky ReLU activation layer before a second 3D CNN, 3D BN, and leaky ReLU follows. Finally, we pass the output tensor of size $(C, F, H, W)$ through a 3D CNN output layer with two kernels of size $(1, 1, 1)$ and stride $(1, 1, 1)$ and apply the final Sigmoid function to get normalized values between 0 and 1. The first output channel maps to a predefined range interval resulting in the future range predictions. As done by Weng et al. [8], the second channel contains a probability for each range image point to be a valid point for re-projection. The re-projection mask keeps all points with a probability above 0.5. This makes it possible to mask out e.g. the sky since there are no ground truth points available.

## 3.2 Skip Connections and Horizontal Circular Padding

We use strided 3D convolutions and downsample the range images as described in Sec. 3.1. The reduced size of the feature space however causes a loss of details in the predicted range images. We address this problem by adding skip connections [32] between the encoder and decoder to maintain details from the input scene. As shown in Fig. 3, the feature maps bypass the remaining encoding steps and the mirrored decoder stage concatenates them with the previously upsampled feature volume along the channel dimension. Concatenation enables the network to account for the temporal offset between encoder and decoder feature maps. A combination of 3D convolutions, 3D batch normalization, and leaky ReLU follows to merge the features back to the original number of channels while maintaining the temporal and spatial dimensions. We investigate the effect of skip connections in Sec. 4.1 and Sec. 4.4 and show that they maintain details in the predicted point clouds.

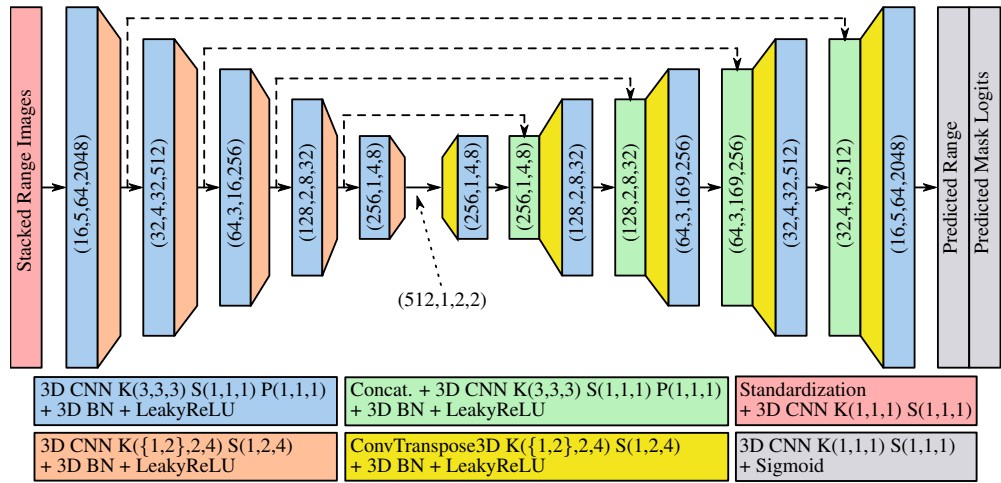

Figure 3: Our detailed spatio-temporal architecture using 3D convolutional neural networks with feature map sizes $(C_i, P_i, H_i, W_i)$. Solid and dashed arrows indicate information flow and skip connections, respectively. Details of each block are in the colorized boxes with kernel sizes K, stride S and padding P. The feature map is not padded in case P is not specified. To be viewed in color.

Another challenge when using range images for 3D point cloud prediction is to maintain spatial consistency on the horizontal borders of the range images. Range images obtained by rotating LiDAR sensors such as a Velodyne or an Ouster sensor, are panoramic images and have strong horizontal correlations between the image borders. If the ego vehicle is rotating along the vertical $z$ axis, an object passing the left border of the range image will appear on the right border. To take this property into account, we introduce circular padding for the width dimension. We pad the left side of each feature map with its right side and vice versa. The vertical dimension is padded with zeros. An experiment on the effectiveness of this padding is provided in Sec. 4.1.

### 3.3 Training

When training the network, we project the ground truth point clouds into the range images of size $(H, W)$, which allows for the computation of 2D image-based losses. We slice the data into samples of sequences consisting of $P$ past and $F$ future frames, where subsequent samples are one frame apart. Our architecture is trained with a combination of multiple losses. The average range loss $\mathcal{L}_{\text{R},t}$ at time step $t$ between a predicted range image $(\hat{r}_{i,j})_t \in \mathbb{R}^{H \times W}$ and a ground truth range image $(r_{i,j})_t \in \mathbb{R}^{H \times W}$ is given by

$$\mathcal{L}_{\text{R},t} = \frac{1}{HW} \sum_{i,j} \Delta r_{i,j}, \quad \text{with} \quad \Delta r_{i,j} = \begin{cases} \|\hat{r}_{i,j} - r_{i,j}\|_1 & \text{, if } r_{i,j} > 0 \\ 0 & \text{, otherwise} \end{cases}, \tag{1}$$

such that the range loss is only computed for valid ground truth points. We train the mask output at time step $t$ with predicted probabilities $(\hat{m}_{i,j})_t \in \mathbb{R}^{H \times W}$ with a binary cross-entropy loss

$$\mathcal{L}_{\text{M},t} = \frac{1}{HW} \sum_{i,j} -y_{i,j} \log(\hat{m}_{i,j}) - (1 - y_{i,j}) \log(1 - \hat{m}_{i,j}), \tag{2}$$

where $y_{i,j}$ is 1 in case the ground truth point is valid and 0 otherwise. Both losses only consider the predicted 2D range images and not the re-projected point clouds to compute the loss faster. For comparison to 3D-based methods like MoNet [9], we fine-tune our model on a 3D point-based loss. A common metric for comparing 3D point clouds is the Chamfer distance [33]. At time $t$, we re-project the masked range image into a 3D point cloud $\hat{\mathcal{S}}_t = \{\hat{\boldsymbol{p}} \in \mathbb{R}^3\}, |\hat{\mathcal{S}}_t| = N$, and compare it with the ground truth point cloud $\mathcal{S}_t = \{\boldsymbol{p} \in \mathbb{R}^3\}, |\mathcal{S}_t| = M$, by computing

$$\mathcal{L}_{\text{CD},t} = \frac{1}{N} \sum_{\hat{\boldsymbol{p}} \in \hat{\mathcal{S}}_t} \min_{\boldsymbol{p} \in \mathcal{S}_t} \|\hat{\boldsymbol{p}} - \boldsymbol{p}\|_2^2 + \frac{1}{M} \sum_{\boldsymbol{p} \in \mathcal{S}_t} \min_{\hat{\boldsymbol{p}} \in \hat{\mathcal{S}}_t} \|\hat{\boldsymbol{p}} - \boldsymbol{p}\|_2^2. \tag{3}$$

The computation of the Chamfer distance is in general slow due to the search for nearest neighbors. A fast to compute image-based loss using $\mathcal{L}_{\text{R},t}$ and $\mathcal{L}_{\text{M},t}$ is about 2.5 times faster compared to a loss

including the 3D Chamfer distance. We propose a training scheme with only range and mask loss for pre-training. This results in a good initialization for fine-tuning including a Chamfer distance loss. We provide an experiment on the training scheme in Sec. 4.4. Given a current time step $t = T$, our total loss function for $F$ future time steps is given by

$$\mathcal{L} = \sum_{t=T+1}^{T+F} \mathcal{L}_{\text{R,t}} + \alpha_{\text{M}}\mathcal{L}_{\text{M,t}} + \alpha_{\text{CD}}\mathcal{L}_{\text{CD,t}}, \tag{4}$$

with tunable weighting parameters $\alpha_{\text{M}}$ and $\alpha_{\text{CD}}$. During the experiments, we experienced that setting $\alpha_{\text{M}} = 1.0$ and $\alpha_{\text{CD}} = 0.0$ for pre-training and $\alpha_{\text{M}} = 1.0$ and $\alpha_{\text{CD}} = 1.0$ for fine-tuning worked best. We train our network parameters with the Adam optimizer [34], a learning rate of $10^{-3}$, and accumulate the gradients for 16 samples. We use an exponential decaying learning rate schedule. All models require less than 50 epochs to converge.

## 4 Experimental Evaluation

In this work, we present an approach for self-supervised point cloud prediction using spatio-temporal 3D convolutional neural networks. We present our experiments to show the capabilities of our method and to support our key claims, which are: (i) predicting a sequence of detailed future 3D point clouds from a given input sequence by a fast joint spatio-temporal point cloud processing using temporal 3D convolutional networks, (ii) outperforming state-of-the-art point cloud prediction approaches, (iii) generalizing well to unseen environments, and operate online faster than a typical rotating 3D LiDAR sensor frame rate of 10 Hz. We follow the experimental setting of MoNet [9] with the KITTI Odometry dataset [35] and predict the next 5 frames from the past 5 LiDAR scans captured at 10 Hz. Please see Appendix B for further details on the experimental setting.

### 4.1 Qualitative Results

The first experiment supports our claim that our model can predict a sequence of detailed future 3D point clouds using a temporal convolutional network and demonstrate the effectiveness of our proposed skip connections and circular padding. The situation in Fig. 4 from test sequence 08 shows an ego-vehicle finishing a left turn while passing a close vehicle. This is a challenging scenario because of the different scales of close and far objects and the ego-motion of the car. Our approach pre-trained on range and mask loss only is capable of predicting the future point clouds across 5 time steps and maintains details in the scene. Without skip connections, small objects are lost due to the compression (green circle). With typical zero padding, the points belonging to the car in the back of the ego vehicle (black circle) are not consistently predicted since they span across the range image borders, while our approach can solve this problem by using the proposed circular padding.

### 4.2 Quantitative Results

To support the claim that our method outperforms state-of-the-art point cloud prediction approaches, we compare the quantitative results to multiple baselines. Note that Lu et al. [9] operate and also evaluate on sub-sampled point clouds, whereas our approach can predict full-size point clouds. To account for this, Tab. 1 (*Left*) shows the performance of our method on sampled point clouds with two different sampling rates used in MoNet [9]. For both samplings, our method achieves a better result for larger prediction steps. However, it can be seen that sampling points from the original point clouds during evaluation strongly affects the Chamfer distance of our full-sized prediction. Thus, we provide an evaluation on full-scale point clouds to demonstrate the true performance of our approach, see Tab. 1 (*Right*). We compare our results to an *Identity* baseline that takes the last received scan as a constant prediction for all future time steps. The *Constant Velocity* and *Ray Tracing* baselines estimate the transformation between the last two received scans with a SLAM approach [36] as a constant velocity prediction of the sensor. At each prediction step, the *Constant Velocity* baseline transforms the last received point cloud according to the predicted motion. The *Ray Tracing* baseline aggregates all five transformed past point clouds to render a more dense range image. Since a lot of points in the scene are static, these are very strong baselines that work well for constant ego-motion. Note that in contrast to our method, all three baselines do not consider points belonging to moving objects. The results in Tab. 1 show that our method outperforms all baselines at larger

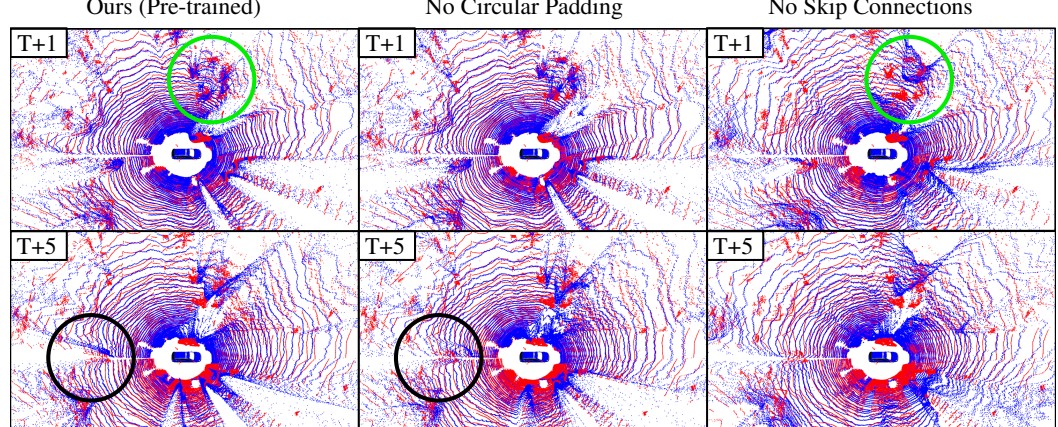

| Ours (Pre-trained) | No Circular Padding | No Skip Connections |

Figure 4: Qualitative comparison of our pre-trained method and two ablated models with predicted (blue) and ground truth (red) point clouds from test sequence 08. Given a past sequence of 5 point clouds at time $T$, the upper row shows the first predicted frame at $T+1$ and the lower row the last predicted frame at $T+5$. The green circle encloses an area where details are lost without skip connections and the points in the black circle demonstrate that circular padding maintains spatial consistency at the range image border. Best viewed in color.

| | Sampled Point Clouds | | | | | Full-scale Point Clouds | | | |
|---|---|---|---|---|---|---|---|---|---|
| Prediction Step | PointLSTM [26] | Scene Flow [9] | MoNet [9] | Ours 32,768 pts | Ours 65,536 pts | Identity Baseline | Const. Vel. Baseline | Ray Tracing Baseline | Ours |
| 1 | 0.332 | 0.503 | **0.278** | 0.367 | 0.288 | 0.271 (0.144) | **0.105** (0.067) | 0.158 (0.095) | 0.254 (0.124) |
| 2 | 0.561 | 0.854 | 0.409 | 0.446 | **0.352** | 0.719 (0.448) | **0.217** (0.221) | 0.253 (0.246) | 0.310 (0.188) |
| 3 | 0.810 | 1.323 | 0.549 | 0.546 | **0.428** | 1.216 (0.798) | 0.385 (0.443) | 0.388 (0.460) | **0.378** (0.262) |
| 4 | 1.054 | 1.913 | 0.692 | 0.638 | **0.509** | 1.727 (1.169) | 0.604 (0.711) | 0.556 (0.719) | **0.448** (0.370) |
| 5 | 1.299 | 2.610 | 0.842 | 0.763 | **0.615** | 2.240 (1.156) | 0.852 (0.968) | 0.751 (0.971) | **0.547** (0.487) |
| Mean (Std) | 0.811 | 1.440 | 0.554 | 0.552 | **0.439** | 1.235 (1.192) | 0.433 (0.641) | 0.421 (0.627) | **0.387** (0.331) |

Table 1: Chamfer distance results on KITTI Odometry test sequences 08 to 10 in $\left[\mathrm{m}^2\right]$ with sampled (*left*) and full-scale (*right*) points clouds. We first train our model on range and mask loss and then fine-tune including a Chamfer distance loss for 10 epochs as described in Sec. 3.3. Best mean results in bold for sampled and full-scale point clouds, respectively. For full-scale point clouds, the standard deviation of the Chamfer distance is given in parentheses. Please see Appendix E for a more detailed statistical analysis.

prediction steps and that it can reason about the sensor's ego-motion and the motion of moving points. The standard deviation of the Chamfer distance indicates that the results of our approach are more consistent throughout the test set whereas the baseline results suffer from outliers in case of moving objects or non-linear ego-motion.

### 4.3 Runtime

For the runtime, our approach takes on average 11 ms (90 Hz) for predicting 5 future point clouds with up to 131,072 points each on a system with an Intel i7-6850K CPU and an Nvidia GeForce RTX 2080 Ti GPU, which is faster than the sensor frame rate, i.e. 10 Hz, of a typical rotating 3D LiDAR sensor and faster than existing approaches like MoNet [9] using the same GPU. Compared to a runtime of 280 ms reported by Lu et al. [9] for MoNet for predicting 5 future point clouds with 65,536 points each, our approach is 25 times faster for twice as many points.

### 4.4 Ablation Study

In this experiment, we provide a more detailed analysis of the effect of the model architecture and the proposed training scheme. We average the metrics over all validation samples and time steps as shown in Tab. 2. Note that we always report the Chamfer distance but it is only used during training if specified. First, we modify the size of the network by decreasing (*Small*) or increasing (*Large*) the number of channels of our model by a factor of 2. The larger model predicts slightly better masked range images but also requires 4 times more parameters. All losses increase without skip connections (*No Skip Connections*). This is because a lot of objects are not predicted due to the large feature compression. The effect of circular padding (*No Circular Padding*) is not visible quantitatively because of the small number of affected points and we refer to Sec. 4.1 for a qualitative evaluation. Finally, we show that our fine-tuned model (*Ours*) achieves similar performance

| Property | Small | Large | No Skip Connections | No Circular Padding | CD Loss From Start | Ours (Pre-trained) | Ours |
|---|---|---|---|---|---|---|---|
| 4 M Parameters | ✗ | | | | | | |
| 17 M Parameters | | | ✗ | ✗ | ✗ | ✗ | ✗ |
| 68 M Parameters | | ✗ | | | | | |
| Skip Connections | ✗ | ✗ | | ✗ | ✗ | ✗ | ✗ |
| Circular Padding | ✗ | ✗ | ✗ | | ✗ | ✗ | ✗ |
| Loss Weight $\alpha_{CD}$ | 0.0 | 0.0 | 0.0 | 0.0 | 1.0 | 0.0 | $0.0 \rightarrow 1.0$ |
| $\mathcal{L}_R$ [m] | 0.867 | 0.741 | 1.258 | 0.801 | 0.805 | 0.798 | 0.858 |
| $\mathcal{L}_{Mask}$ | 0.309 | 0.288 | 0.337 | 0.297 | 0.300 | 0.299 | 0.301 |
| $\mathcal{L}_{CD}$ $[m^2]$ | 1.110 | 0.714 | 2.759 | 0.885 | 0.487 | 0.985 | 0.480 |

Table 2: Evaluation of models with varying architectures, padding, and training schemes on the KITTI Odometry validation sequences 06 and 07. Each variation (columns) is given by a combination of different design decisions (rows) indicated by crosses (✗), e.g., variation "Large" is a model with 68 M parameters using skip connections and circular padding. The arrow ($\rightarrow$) indicates a pre-training and fine-tuning with two different loss weights $\alpha_{CD}$ as discussed in Sec. 3.3.

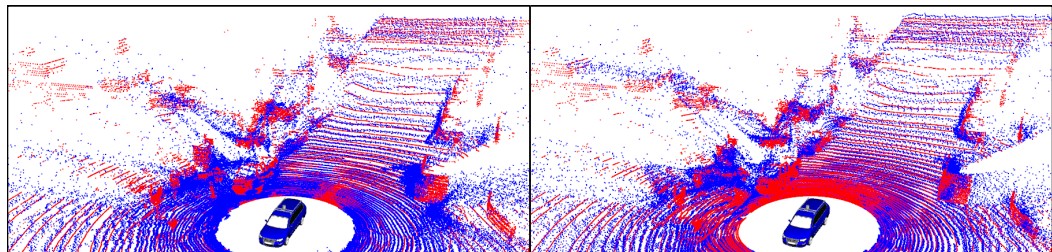

Figure 5: Predicted (blue) and ground truth (red) future point clouds on the Apollo-SouthBay Columbia-Park [37] test set at prediction step 5. *Left*: Model only trained on KITTI odometry data. *Right*: After training one epoch on the Apollo training set.

on the validation set compared to a model optimized with the Chamfer distance loss from the beginning (*CD Loss From Start*). Our proposed training scheme with pre-training on fast-to-compute image-based losses gives a good pre-trained model for fine-tuning with the 3D Chamfer distance loss while reducing training time.

## 4.5  Generalization

Finally, we aim at supporting our claim that our method generalizes well to new, unseen environments. We first evaluate a model fine-tuned on KITTI Odometry on the Apollo-SouthBay ColumbiaPark [37] test set. Note that this data is from an unseen, *different* environment (collected in the U.S.) with a *different* type of car setup, comparing to the training data (collected in Germany). Our model achieves a mean Chamfer distance of $0.934\,\mathrm{m}^2$ on the Apollo test set. The performance gap compared to the results in Tab. 1 is due to the fact that the training data has a maximum range of $85\,\mathrm{m}$ but the Apollo test data contains points up to a range of $110\,\mathrm{m}$. Thus, we fine-tune the model on the Apollo-SouthBay ColumbiaPark [37] training set for a single epoch. This yields a mean Chamfer distance of $0.426\,\mathrm{m}^2$ and demonstrates the generalization capability of our approach and the advantage of self-supervised training. We provide an additional experiment on the effect of using more training data in Appendix C. Fig. 5 shows that the fine-tuned model reduces the Chamfer distance by predicting more accurate shapes and more distant points. Note that compared to Fig. 4, the different sensor mount causes a changed pattern of invalid points around the car. Our approach can correctly predict the re-projection mask.

## 5  Conclusion

In this paper, we presented a novel approach to self-supervised point cloud prediction with 3D convolutional neural networks. Our method exploits the range image representation to jointly extract spatio-temporal features from the input point clouds sequence. This allows us to successfully predict detailed, full-scale, future 3D point clouds during online operation. We implemented and evaluated our approach on different datasets and provided comparisons to other existing techniques and supported all claims made in this work. The experiments suggest that our method outperforms existing baselines and generalizes well to different sensor mounts in unseen environments.

**Acknowledgments**

We thank Lu et al. [9] for providing implementation details and experimental results.

This work has partially been funded by the Deutsche Forschungsgemeinschaft (DFG, German Research Foundation) under Germany's Excellence Strategy, EXC-2070 – 390732324 – PhenoRob, by the European Union's Horizon 2020 research and innovation programme under grant agreement No 101017008 (Harmony), and by the Federal Ministry of Food and Agriculture (BMEL) based on a decision of the Parliament of the Federal Republic of Germany via the Federal Office for Agriculture and Food (BLE) under the innovation support programme under funding no 28DK108B20 (RegisTer).

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
