# OpenReview forum: "Self-supervised Point Cloud Prediction Using 3D Spatio-temporal Convolutional Networks"
_robot-learning.org/CoRL/2021/Conference — CoRL2021 Poster_

### Official Review · Reviewer_n5qa · 2021-07-09

**Originality:** Very Good
**Technical Quality:** Excellent
**Clarity Of Presentation:** Excellent
**Impact:** 4

**Recommendation:**

Weak Accept: I recommend accepting the paper, but will not argue for my recommendation if the majority of other reviewers have a different opinion.

**Summary:**

The paper proposes a self-supervised method for predicting future LiDaR point cloud measurements, using a encoder-decoder type neural network. Data from past point cloud frames is first converted into 2D range images, stacked along the time dimension to obtain a 3D tensor. This tensor is an input to an 3D convolutional encoder network. The latent representation is upsampled by the decoder module to produce a stack of predicted future range images. Skip connections from intermediate layers of the encoder to the decoder are added to maintain spatial and temporal details. The method is pre-trained using fast to compute image-based losses, and fine-tuned using the Chamfer loss on the predicted point clouds. Empirically, the proposed method performs strongly compared to state of the art, with ablation studies backing up the decision choices. Finally, the generalization capabilities of the method to new environments are demonstrated.

**Issues:**

I would be happy to have comments from the authors on the limitations I mentioned, and hope the minor corrections can be implemented.

**Reviewer Expertise:**

Good: General knowledge of the area

**Strengths And Weaknesses:**

Strengths:

- The method is technically solid, with many interesting innovations described in dealing with point cloud data as range images, processing them using 3D CNNs (esp. skip connections and circular padding), and the self-supervised training method with pre-training and fine-tuning phases. Empirically, the outcome is a comparably fast system that is simple to train. I expect that this work will have a significant impact in this area of research.

- The empirical results are stronger than current state of the art, especially for longer prediction steps. The ablation study in Sect. 4.4. clearly shows the usefulness of the different design decisions, in particular that the skip connections and circular padding are very helpful.

- The paper is overall very clearly written and presented, and an enjoyable read.

Limitations:

- Even if prediction accuracy measured by Chamfer distance are better for the proposed method, it is not clear if this necessarily translates to improved performance in downstream tasks. The results of this paper could be further strengthened by demonstrating performance improvements over other methods on some downstream tasks using the predicted point clouds.

- While the ablation studies are sufficient, it could have been nice to explore some further aspects. Longer prediction horizons might be interesting, as the current 5 steps corresponds to 0.5 seconds only, as far as I understand. Sensitivity to noise or other disturbances in the input data could be explored.

- The generalization study is nice, but the results here are not very thorough. This aspect might be interesting to explore more deeply in future work.


Other comments/suggestions/minor:
- Sect. 4.1. could be moved to after the main ablation results. I think this would be clearer, showing qualitative examples after demonstrating the usefulness of the two additions quantitatively.
- Fig 2 caption: rang --> range

**Summary Of Recommendation:**

This paper presents a strong contribution for point cloud prediction. The empirical study is thorough, demonstrating all necessary aspects of the method. While there are shortcomings, it can be argued that they will be addressed in follow-up works. My only reason for a weak acceptance recommendation only is that I lack the expertise in this particular area to take a stronger stand.

---

> ### Author Response · Authors · 2021-08-25
> **Response to Reviewer n5qa**
>
> # General
> Thank you for your constructive feedback. We comment on the limitations and minor suggestions below. Implemented corrections are highlighted in blue in the revised version.
>
> ---
>
> # Limitations:
> *The results of this paper could be further strengthened by demonstrating performance improvements over other methods on some downstream tasks using the predicted point clouds.*
> >This is a great suggestion and we would like to apply our method to other downstream tasks in the future. Regarding the limited time and paper space, we decide to put this to future work.
>
> *While the ablation studies are sufficient, it could have been nice to explore some further aspects. Longer prediction horizons might be interesting, as the current 5 steps correspond to 0.5 seconds only, as far as I understand. Sensitivity to noise or other disturbances in the input data could be explored.*
> >We refer to our answer to reviewer 2bTD for extending the method to longer prediction horizons. Evaluating the sensitivity to noise and disturbances is indeed an interesting direction we might take in the future.
>
> *The generalization study is nice, but the results here are not very thorough. This aspect might be interesting to explore more deeply in future work.*
> >We agree that generalization is an important experiment, especially since the method is self-supervised and does not require labeled data. Due to the space constraints, we already moved one experiment on the amount of training data to the supplementary material, see Appendix C. We plan to evaluate the approach on other datasets with more dynamic scenes in the future.
>
> ---
>
> # Minor
> *Sect. 4.1. could be moved to after the main ablation results. I think this would be clearer, showing qualitative examples after demonstrating the usefulness of the two additions quantitatively.*
> >We acknowledge the reviewer’s intention to first show quantitative results and then qualitative examples. We decided to keep the order of experiments to be in line with the claims made in the paper. Since our first claim is to have developed a new method to predict a sequence of future point clouds using spatio-temporal 3D convolutional networks, we think the reader should first get an intuition of how the predictions look like which is easier than interpreting the Chamfer distance metric in the quantitative analysis.
>
> *Fig 2 caption: rang --> range*
> >We changed this in the revised version.

---

> > ### Comment · Reviewer_n5qa · 2021-08-31
> > **thank you**
> >
> > Thank you for the response. I remain by my original positive rating and hope to see future work addressing the remaining concerns.

---

### Official Review · Reviewer_2bTD · 2021-07-25

**Originality:** Good
**Technical Quality:** Good
**Clarity Of Presentation:** Good
**Impact:** 3

**Recommendation:**

Weak Accept: I recommend accepting the paper, but will not argue for my recommendation if the majority of other reviewers have a different opinion.

**Summary:**

The paper considers the problem of predicting future 3D LiDAR point clouds. It proposes a model that first encodes a single 3D LiDAR scan into a 2D representation, then forms a 3D tensor by concatenating the 2D representations across time, and applies a 3D CNN encoder-decoder architecture. The training does not require any labeled data, but just LiDAR scans across time.

**Issues:**

1. Figure 1 does not show any perceptible change in the predicted cloud. The authors can think of showing this in another way?

2. The 3D LiDAR scan to 2D representation is a critical part of the architecture and should be included in the main part of the paper. It is not possible to follow the paper without it.

3. The notation P H W is not explaining in the main paper, the first time it is used.

4. The method is used to predict only the next 5 frames (about 0.5 seconds) from the past 5 LiDAR scans (about 0.5 seconds). This seems like a very short time span for the KITTI dataset. It would have been good if the paper had discussed more the limitations of the model towards extending the prediction horizon.


**Reviewer Expertise:**

Very good: Comprehensive knowledge of the area

**Strengths And Weaknesses:**

Strengths: The proposed idea is simple and is reported to perform better than existing approaches for point cloud prediction. The proposed model can predict faster than the LiDAR sensor frame rate of 10 Hz, which is a meaningful consideration when it comes to practicability of the proposed solution.


**Summary Of Recommendation:**

The paper considers the problem of predicting future 3D LiDAR point clouds and proposes a very simple architecture. The proposed model is reported to outperform some of the methods in the literature. However, the limitations occur on two counts:

(1) The paper does not discuss how far it can predict. Would the model complexity increase if it is asked to predict for the future 100 frames, instead of just 5? How would that compare with the other models - that for example use LSTM (as LSTM can track and predict over longer time horizons)? How would this impact the time required to predict - would it still be within 10 Hz?

(2) The proposed model seems like a simple application of an existing 3D convolutional encoder-decoder architecture (with slight modification). There isn't much novelty.

---

> ### Author Response · Authors · 2021-08-25
> **Response to Reviewer 2bTD**
>
> # General
> Thank you for your constructive feedback. We address both mentioned limitations and all four major issues below. Changes are highlighted in blue in the revised version.
>
> ---
>
> # Limitations
> *The paper does not discuss how far it can predict. Would the model complexity increase if it is asked to predict for the future 100 frames, instead of just 5? How would that compare with the other models - that for example use LSTM (as LSTM can track and predict over longer time horizons)? How would this impact the time required to predict - would it still be within 10 Hz?*
> >There are two ways to increase the prediction horizon:
> >
> >1. The number of predicted frames is determined by the number of kernels of the last output layer and could be set to a larger value before training. This slightly increases the complexity and prediction time. One challenge we see here is the required memory for training when computing the loss for 100 predicted frames.
> >
> >2. It is also possible to extend the prediction horizon in an autoregressive manner by feeding back the predicted range images as the next input tensor. This has been explored for video prediction by Aigner et al. [25]. This does not increase the model’s complexity but increases the time needed for prediction. Since LSTM-based methods by nature decode the output sequentially, we expect our method still to be faster. For predicting 100 frames, our method needs 20 sequential predictions of 5 future frames each at 90 Hz resulting in an effective frequency of 4.5 Hz. MoNet [9] processes and predicts 5 downsampled point clouds sequentially at a total frequency of 3.6 Hz, which reduces to an effective frequency of 0.18 Hz for predicting 100 points clouds. This frequency is far below our achieved frequency and only considers downsampled point clouds. In general, we focused on a 5 frame prediction to match the setup of MoNet [9] for comparison. We agree with the reviewer and believe that longer time horizons should be investigated in future work. However, we question the necessity to predict a large time horizon like 100 frames, since for long-term predictions additional information like road structure, traffic rules, and indicators like turn signals should be considered. We added an explanation of how to use the method for predicting a longer time horizon in Sec. 3.1.
>
> *The proposed model seems like a simple application of an existing 3D convolutional encoder-decoder architecture (with slight modification). There isn't much novelty.*
> >The proposed architecture is designed and implemented by us from scratch. Our work is for example different from the 3D CNN-based video prediction method FutureGAN [25] since we implement a multi-head prediction of mask and range images. We also add and evaluate skip connections and circular padding in this context and therefore claim novelty in using this new architecture for point cloud prediction with the proposed pre-training and fine-tuning phases.
>
> ---
>
> # Issues
>
> *1. Figure 1 does not show any perceptible change in the predicted cloud. The authors can think of showing this in another way?*
> >We modified Figure 1 and show the first and last frame of the input and output sequence, respectively. With this, the temporal change in the point clouds is better visible.
>
> *2. The 3D LiDAR scan to 2D representation is a critical part of the architecture and should be included in the main part of the paper.*
> >Unfortunately, there is not enough space left in the main paper. Since the equations are commonly used and not part of our contribution, we slightly modified Sec. 3.1 in the main paper and kept the equations and further details in Appendix A.
>
> *3. The notation P H W is not explaining in the main paper, the first time it is used.*
> >We now define P, H, and W in Sec. 3.1.
>
> *4. It would have been good if the paper had discussed more the limitations of the model towards extending the prediction horizon.*
> >We discussed the extension of the prediction horizon above and modified the description of the decoder in Sec. 3.1.
>
> ---
>
> # References
> [9] F. Lu, G. Chen, Y. Liu, Z. Li, S. Qu, and T. Zou. MoNet: Motion-based Point Cloud Prediction Network. arXiv preprint, abs/2011.10812, 2020.
>
> [25] S. Aigner and M. Korner. FutureGAN: Anticipating the Future Frames of VideoSequences using Spatio-Temporal 3d Convolutions in Progressively Growing GANs. In Proc. of the IEEE/CVF Conf. on Computer Vision and Pattern Recognition (CVPR), 2018.

---

### Official Review · Reviewer_puBM · 2021-07-27

**Originality:** Good
**Technical Quality:** Good
**Clarity Of Presentation:** Very Good
**Impact:** 3

**Recommendation:**

Weak Accept: I recommend accepting the paper, but will not argue for my recommendation if the majority of other reviewers have a different opinion.

**Summary:**

The paper proposes a range image based encoder-decoder architecture to predict future pointclouds. A temporal stack of range images is used to predict another stack of future range images. The experiments evaluate the performance of the architecture in both quality and computational speed. The impact of the different architecture's components is also evaluated.

**Issues:**

- Lack of statistical evaluation of the results
- Could benefit from a more thorough discussion of architecture design choices

**Reviewer Expertise:**

Good: General knowledge of the area

**Strengths And Weaknesses:**

The proposed method uses common ideas and techniques from image based deep learning with typical range image representations. While the components used are not novel per se, the execution and description of the approach are easy to read and follow, with the method itself being technically sound.

The most prominent aspect that could be improved in the method description is to provide more insight and discussion into why specific aspects are used. For example, the method uses simple convolutions to capture temporal aspects via the "depth" of the image stack instead of LSTM or other mechanisms. What are the expected benefits and drawbacks of this? The encoder-decoder structure is also not exactly the typical reconstruction focused one, as the reconstructed image is not the same as the input. What is the reason this architecture was chosen over other plausible alternatives, such as a pure reconstruction encoder-decoder whose low-dimensional embedding is then used for future prediction via an MLP architecture? One aspect that was not fully clear is whether or not the receptive field in "time" is different from the spatial one.

The losses used seem sensible, and the explanation for their use is clear. One common issue with such compound losses is that the different components require scaling or tuning with respect to each other. Based on the description in the paper, this is not the case here. Is there a particular reason, and were any tests done to see how sensitive the results are regarding such weighting?

The experiments overall are well done and address questions that are pertinent to the method. My biggest complaint would be the lack of proper statistical information, such as standard deviation or better quantiles. Having this information would provide greater insight into the range of outcomes the various methods produce instead of a singular value such as the mean that is easily affected by outliers. The ablation study is welcome and informative, however also somewhat confusing. The methods labelled "CD Loss From Start" and "Ours" are identical based on their feature sets but have different performances. From the text, I could not decipher the difference between them as the names used do not line up with the headings in the table.

While the paper suggests that the qualitative results shown in Figure 4 showcase the improved performance of the proposed method, I cannot see that. To me, all images look equally "bad" regarding their ability to reconstruct the scan. This is likely due to a 3D point cloud being projected onto a 2D image, making them significantly harder to visualize. I do not doubt that the differences exist, but rather the visualization does not convey this convincingly enough.

Finally, one of the motivations at the beginning of the paper was predicting point clouds for tracking dynamic objects. In light of this, it would have been perfect to evaluate the predictive quality for such a task or at least the quality of the prediction for objects of interest, as opposed to the entire point cloud.

**Summary Of Recommendation:**

Overall the paper presents a sound method that achieves the goals outlined. The approach uses well known and understood components to achieve this but could benefit from a more thorough discussion of these choices. The experiments demonstrate the performance of the method adequately. However, they could be improved by providing a proper statistical evaluation of the results.

---

> ### Author Response · Authors · 2021-08-25
> **Response to Reviewer puBM (1/2)**
>
> # General
> Thank you for your constructive feedback. We address the mentioned weaknesses and both major issues below. We also made appropriate changes in the paper which are highlighted in blue in the revised version.
>
> ---
>
> # Weaknesses
>
> *The most prominent aspect that could be improved in the method description is to provide more insight and discussion into why specific aspects are used. For example, the method uses simple convolutions to capture temporal aspects via the "depth" of the image stack instead of LSTM or other mechanisms. What are the expected benefits and drawbacks of this?*
> >The expected benefits of our proposed spatio-temporal 3D CNN architecture are easier training and faster inference. In contrast to LSTM-based methods that process each scan sequentially and require maintaining a meaningful hidden state vector, our method jointly encodes the input sequences into a latent space followed by a decoder. This makes our approach faster compared to 3D point-based methods using LSTMs like MoNet as discussed in our paper.
> >
> >When operating on range images with ~100.000 pixels each, using a vanilla LSTM requires a preceding encoding of the range images to reduce the feature size. The number of trainable parameters of an LSTM increases quadratically with the number of input features. Weng et al. [8] use CNNs to compress each range image into a small feature vector of size 1024. However, downsampling each range image before feeding it to the LSTM loses details in the scene. Using our convolution-based encoder-decoder architecture, we can use skip connections to preserve details in the scene. Weng et al. [8] discuss the problem of lost details in a recent research talk, see [here](https://youtu.be/Z9_hu30lDy8?t=999).
> >
> >A drawback of only using CNNs is that the number of input and output frames used for prediction is fixed. We address this limitation in our comment to reviewer 2bTD and propose a possible solution.
> >
> >We added some explanations in Sec. to relate our architecture to existing LSTM-based methods.
>
> *The encoder-decoder structure is also not exactly the typical reconstruction focused one, as the reconstructed image is not the same as the input. What is the reason this architecture was chosen over other plausible alternatives, such as a pure reconstruction encoder-decoder whose low-dimensional embedding is then used for future prediction via an MLP architecture?*
> >Since an MLP works on one-dimensional data and does not consider spatial and temporal dimensions, we decided to use a convolutional architecture to process local patterns in space and time. For CNNs, the parameters are shared in the convolutional kernels and are independent of the absolute position in the input tensor. The resulting network requires less trainable weights and the risk of overfitting is reduced compared to an MLP architecture. We added some arguments for our architecture choice to Sec. 3.1.
>
> *One aspect that was not fully clear is whether or not the receptive field in "time" is different from the spatial one.*
> >In general, the size of the receptive field depends on the number of layers and convolutional hyperparameters like kernel size, dilation rate, and padding. The spatial receptive field needs to be large enough to capture the motion of an object from the positions in the past range images to the positions in the future range images. The temporal receptive field should span along the full temporal dimension. This ensures that all past range images are taken into account for each predicted future range image. We achieve this by compressing the time dimension of the feature map into one, therefore each output time step depends on inputs from all past time steps. We rephrased the sentence in Sec. 3.1 to make clear that there is a receptive field for each dimension.
>
> *Based on the description in the paper, this is not the case here. Is there a particular reason, and were any tests done to see how sensitive the results are regarding such weighting?*
> >We experimented with different weights and found that the Chamfer distance metric on the validation set is not sensitive to this parameter. We chose the weights for the range and mask loss to be 1 and the Chamfer distance loss weight to be 0 for pre-training and 1 for fine-tuning, therefore they have been omitted in the initial submission. We now added explicit weight variables to equation (4) and clarified it in the paper.

---

> > ### Author Response · Authors · 2021-08-25
> > **Response to Reviewer puBM (2/2)**
> >
> > *My biggest complaint would be the lack of proper statistical information, such as standard deviation or better quantiles. Having this information would provide greater insight into the range of outcomes the various methods produce instead of a singular value such as the mean that is easily affected by outliers.*
> > >Since the raw results are not available in the case of MoNet [9], we now added a statistical analysis to the quantitative results for full-scale point clouds. The revised paper contains the standard deviations of the per-step Chamfer distances as well as the standard deviation across all steps. Due to space constraints, we provide detailed box plots with median, minimum, maximum, first quartile, and third quartile of the baselines as well as our method in Appendix E. Overall, the statistical analysis shows that the performance of our method is less affected by outliers and outperforms the baselines for larger prediction horizons. The deviation of the baseline results can be explained by the fact that they ignore moving objects. Since not all LiDAR frames in the KITTI Odometry dataset contain moving objects, the constant velocity baseline is strong at predicting static scenes but fails when points move relative to the ego-motion or when the static scene changes due to a non-linear ego-motion.
> >
> > *The methods labelled "CD Loss From Start" and "Ours" are identical based on their feature sets but have different performances*
> > >We modified the description of the feature set in Table 2 and made the methods’ names consistent throughout the table and text.
> >
> > *To me, all images look equally "bad" regarding their ability to reconstruct the scan. This is likely due to a 3D point cloud being projected onto a 2D image, making them significantly harder to visualize.*
> > >We would like to point out that the task is not to reconstruct the scan but to predict an unknown future scan which is not a trivial task. Therefore, the quality of a predicted future point cloud is expected to be worse than its ground truth counterpart. We show that without using skip connections or without circular padding, some future points are not predicted at all as indicated by the colorized circles.
> >
> > *Finally, one of the motivations at the beginning of the paper was predicting point clouds for tracking dynamic objects. In light of this, it would have been perfect to evaluate the predictive quality for such a task or at least the quality of the prediction for objects of interest, as opposed to the entire point cloud.*
> > >We agree with the reviewer that a task-specific evaluation is very interesting and should be the next step. With respect to the limited time and paper space, we shift this evaluation to future work.
> >
> > ---
> >
> > # Issues
> > *Lack of statistical evaluation of the results*
> > >This issue has been addressed by our comment above and by making appropriate changes in the paper, see Table 1 and Appendix E.
> >
> > *Could benefit from a more thorough discussion of architecture design choices*
> > >We modified Sec. 2 to emphasize the benefits of using a convolution-based architecture compared to LSTM-based approaches. Additionally, we now discuss the use of an MLP architecture in Sec. 3.1. Due to the limited space in the main paper, we will use the comment section to give more detailed answers to the questions raised by the reviewer.
> >
> > ---
> >
> > # References
> >
> > [9] F. Lu, G. Chen, Y. Liu, Z. Li, S. Qu, and T. Zou. MoNet: Motion-based Point Cloud Prediction Network. arXiv preprint, abs/2011.10812, 2020.
> >
> > [8] X. Weng, J. Wang, S. Levine, K. Kitani, and N. Rhinehart. Inverting the Pose Forecasting Pipeline with SPF2: Sequential Pointcloud Forecasting for Sequential Pose Forecasting. In Proc. of the Conf. on Robot Learning (CoRL), 2020.

---

> > > ### Comment · Reviewer_puBM · 2021-08-31
> > > **Thank you**
> > >
> > > Thank you for the detailed clarifications and the additions to the paper. Based on the standard deviation and quantiles you show it seems that the larger the horizon the bigger of an advantage your method has. This might be interesting to look at in the future.

---

### Meta-Review · Area_Chair_RKce · 2021-08-12

**Recommendation:** Accept (Poster)
**Confidence:** 4

**Metareview:**

The reviews are generally positive. The AC endorses the reviewers' recommendation. The authors are encouraged to address questions and concerns raised in the reviews.

---

> ### Author Response · Authors · 2021-08-25
> **Response to Area Chair RKce**
>
> We thank the area chair and the reviewers for their constructive feedback. We address all questions and concerns below and uploaded a revised version of the paper with all changes based on the reviewers’ recommendations highlighted in blue. For the final version, we will remove the blue color.

---

### Decision · Program_Chairs · 2021-09-13

**Decision:**

Accept (Poster)

**Comment:**

The reviews are generally positive. The AC endorses the reviewers' recommendation. The authors are encouraged to address questions and concerns raised in the reviews.